The complete mitochondrial genome and description of a new cryptic Brazilian species of Metopiellus Raffray (Coleoptera: Staphylinidae: Pselaphinae)

Asenjo Angélico pukara8@yahoo.com 1
de Oliveira Marcus Paulo Alves 2
Oliveira Renato R.M. 1 3
Pires Eder Soares 1
Valois Marcely 1
Oliveira Guilherme 1
Vasconcelos Santelmo santelmo.vasconcelos@itv.org 1
1 Instituto Tecnológico Vale , Belém , Pará , Brazil
2 BioEspeleo Consultoria Ambiental , Lavras , Minas Gerais , Brazil
3 Universidade Federal de Minas Gerais , Belo Horizonte , Minas Gerais , Brazil
Mitchell Andrew
Electronic publication date: 2023 Jul 27
Publication date: 2023
Volume: 11
Electronic Location ID: e15697
Received 2023 Mar 23; Accepted 2023 Jun 14
Copyright: ©2023 Asenjo et al.
Copyright year: 2023
Copyright holder: Asenjo et al.
License: This is an open access article distributed under the terms of the Creative Commons Attribution License, which permits unrestricted use, distribution, reproduction and adaptation in any medium and for any purpose provided that it is properly attributed. For attribution, the original author(s), title, publication source (PeerJ) and either DOI or URL of the article must be cited.
License URL: https://creativecommons.org/licenses/by/4.0/

Keywords: Amazon basin, Beetle, Brazil, Pselaphinae, Taxonomy

Funding: Vale S.A. (Projeto Diversidade Biológica de Cavernas, R100603.CD.0X; Projeto Centro de Triagem de Invertebrados, R100603.CT.0X) Guilherme Oliveira is a CNPq (Conselho Nacional de Desenvolvimento Científico) fellow 307479/2016-1 CNPq (Conselho Nacional de Desenvolvimento Científico) fellow 307479/2016-1 CNPq 444227/2018-0 402756/2018-5 307479/2016-1 CABANA project RCUK/BB/P027849/1 This work was funded by Vale S.A. (Projeto Diversidade Biológica de Cavernas, R100603.CD.0X; Projeto Centro de Triagem de Invertebrados, R100603.CT.0X). Guilherme Oliveira is a CNPq (Conselho Nacional de Desenvolvimento Científico) fellow (307479/2016-1) and is funded by CNPq (444227/2018-0, 402756/2018-5, 307479/2016-1) and the CABANA project (RCUK/BB/P027849/1). The funders had no role in study design, data collection and analysis, decision to publish, or preparation of the manuscript.

==============================
Metopiellus Raffray, 1908 is a genus of South American rove beetles typically found in tropical humid forests. Here we describe a new cryptic species from Eastern Amazon, in northern Brazil, Metopiellus crypticus Asenjo sp. nov., and its major morphologic diagnostic features, which were photographed and illustrated. In addition, we bring the complete mitochondrial genome sequence of M. crypticus sp. nov., and its position within the phylogenetic context of the family, including previously available mitogenomes of Staphylinidae species.

Introduction

All species of the genus Metopiellus are distributed from Colombia to the North of Argentina (Asenjo et al., 2019; Fiorentino, Tocora & Ramirez, 2022). To date, species of the genus were recorded in the Colombian Amazon (M. guanano Fiorentino, Tocora & Ramirez, 2022), three species in the Brazilian Atlantic Forest (M. aglenus Reitter, 1885, M. hirtus Reitter, 1885, and M. painensis Asenjo, Ferreira & Zampaulo Rde, 2017), and one in Argentina (M. sylvaticus Bruch, 1933) (Asenjo et al., 2013; Asenjo, Ferreira & Zampaulo Rde, 2017; Fiorentino, Tocora & Ramirez, 2022). Members of Metopiellus are usually found inhabiting humid microenvironments on the forest floor consisting in decaying plant parts and their possible association with social insects as ants continue to be uncertain (Reitter, 1885; Park, 1942; Wasmann, 1894; Bruch, 1933). Asenjo, Ferreira & Zampaulo Rde (2017) found M. painensis inside the Loca dos Negros II and the Cerâmica caves in southeastern Brazil. This latter species was the only troglobitic Pselaphinae recorded from Brazil, up to date (Asenjo, Ferreira & Zampaulo Rde, 2017).

The aim of this study was to describe a new species from Brazil that was collected in the state of Pará, northern Brazil. The new species was found inhabiting forest areas, similar to other species in the genus, but it was also found in savanna-like environments. We described, for the first time, the complete mitochondrial genome of Metopiellus crypticus Asenjo sp. nov. positioning the new species in the phylogenetic context of Staphylinidae.

Materials & Methods

Field collection and sequencing

A total of 12 specimens of Metopiellus crypticus Asenjo sp. nov. were collected in forest areas from the Serra dos Carajás in Pará, Brazil, in which the individuals were more abundant, and only two specimens from the canga, a savanna-like environment from the region (Figs. 1B–1C), according to the sampling permit 49.994, granted by ICMBio/MMA. Various collection methods were used in all sites (hand collection, litter sampling and hay-bait traps and soil sampling by flotation), but specimens were found only in litter-associated methods (hay-bait traps and soil sampling by flotation). Therefore, it is likely that the edaphic environment is characterized as a preferred habitat for populations of Metopiellus crypticus Asenjo sp. nov. The specimens were immediately fixed in 99% ethanol within a 2 mL centrifuge tube and transported to the laboratory.

Figure 1 Geographic location of collected specimens of Metopiellus crypticus sp. nov.

Brazil and Pará state are in orange, Carajás mineral Province in red and the specific type locality indicated by a white star (A); exact locations where the specimens were collected in the forest ground (yellow dots) and savanna regions (red dots) of Serra Leste (B) (Map data: Google, ©2022 CNES/Airbus, Maxar Technologies); panoramic view of the vegetation in Serra Leste, the savanna vegetations are in the flat area on mountain tops, and forest vegetation on the slopes and valley (picture by Alan Calux) (C).

Total genomic DNA was extracted from three specimens from the type population with the DNeasy Blood & Tissue kit (Qiagen), following the manufacturer’s protocol for insect samples, being deposited at the DNA bank of the Instituto Tecnológico Vale (ITV) under the accession numbers ITV10661, ITV21026 and ITV21027. Paired-end libraries were constructed from ∼50 ng of genomic DNA using the QXT SureSelect kit (Agilent Technologies, Santa Clara, CA, USA), with which the DNA samples were subjected to an enzymatic random fragmentation and simultaneously bound to adapters, following the manufacturer’s instructions. Then, the samples were purified and subjected to an amplification reaction using primers complementary to the adapters. Afterwards, the libraries were quantified using a Qubit 3.0 (Invitrogen, Waltham, MA, USA) fluorimeter and checked for fragment sizes in a 2100 Bioanalyzer (Agilent Technologies). Finally, the libraries were diluted in a solution of 0.1% Tris–HCl and Tween and pooled to be sequenced in an Illumina NextSeq 500 with the high-output v2 kit (300 cycles, 2 × 150 bp). Resulting raw sequencing reads with base quality <Phred 20 and length <70 bp were trimmed with AdapterRemoval v.2 (Schubert, Lindgreen & Orlando, 2016), resulting in 21,140,835 (ITV10661), 48,275,325 (ITV21026) and 49,851,478 (ITV21027) high quality pairs of reads, which were used to assemble the mitochondrial genomes using NovoPlasty 3.6 (Dierckxsens, Mardulyn & Smits, 2017). A bash script containing the commands for the cleaning and assembling steps is available as Data S1. Finally, the annotations were performed with MITOS2 (Bernt et al., 2013), with subsequent minor manual corrections using Geneious Prime 2021 (Biomatters), by comparing the annotated mitogenomes with previously available data for Pselaphinae species.

Morphological study

Specimens. The apical segments were cleared in a double boiler using 10% KOH during three minutes. Dissections were made under a Leica S8APO (16×–128×) stereo-microscope. Pictures were obtained using the AxioCam 506 color camera connected to an Axio ZoomV16 (ZEISS) stereo microscope and Photoshop CC 2021 was used for image processing, with final plates being assembled in Adobe Illustrator CC 2021. Morphological character terminology, including foveation and its abbreviation followed Chandler (2001). All measurements were made using the Leica S8APO (16×–128×) stereo-microscope, and the width/length ratios were acquired using the widest and longest parts of the respective structures, being presented in millimeters, based on the holotype. We also performed additional measurements using the paratypes, comparing with the data obtained for the holotype (Data S2).

Measurements symbols:

BL body length (from margin of prolongation of head to tergite IX posterior margin)

BW body width (maximum width of elytra)

EL elytral length (maximum)

EW elytral width (maximum)

HL head length (from anterior margin of prolongation of head to head disc posterior margin)

HW head width (maximum)

NW neck width (minimum)

PL pronotum length (maximum)

PW pronotum width (maximum)

In the type label data, quotation marks (“ ”) separate different labels and a slash (/) separates different lines within a label. Text within square brackets [] is explanatory and is not included in the original labels.

Depositories. The specimens examined in this revision are deposited in the following collections (curators in parenthesis):

CEMT - Setor de Entomologia da Coleção Zoológica da Universidade Federal de Mato Grosso, Departamento de Biologia e Zoologia, Cuiabá, Mato Grosso, Brazil (Fernando Vaz-de-Mello).

ISLA - Coleção de Invertebrados Subterrâneos de Lavras, Setor de Zoologia, Departamento de Biologia, Universidade Federal de Lavras, Lavras, Minas Gerais, Brazil (Rodrigo Lopes Ferreira).

ITV - Coleção de DNA do Instituto Tecnológico Vale, Belém, Pará, Brazil (Santelmo Vasconcelos).

MPEG - Museu Paraense Emilio Goeldi, Belém, Pará, Brazil (Orlando Tobias Silveira).

Nomenclatural acts

The electronic version of this article in Portable Document Format (PDF) will represent a published work according to the International Commission on Zoological Nomenclature (ICZN), and hence the new names contained in the electronic version are effectively published under that Code from the electronic edition alone. This published work and the nomenclatural acts it contains have been registered in ZooBank, the online registration system for the ICZN. The ZooBank LSIDs (Life Science Identifiers) can be resolved, and the associated information viewed through any standard web browser, by appending the LSID to the prefix http://zoobank.org/. The LSID for this publication is:

urn:lsid:zoobank.org:pub:85F589F2-AECD-4D52-B545-6363C4CE8E18

The online version of this work is archived and available from the following digital repositories: PeerJ, PubMed Central and CLOCKSS.

Results

Description

Family Staphylinidae Latreille, 1802	
Subfamily Pselaphinae Latreille, 1802	
Tribe Metopiasini Raffray, 1904	
Subtribe Metopiasina Raffray, 1904	
Genus MetopiellusRaffray, 1908	

Metopiellus crypticus Asenjo, new species

(Figs. 1, 2, 3 and 4)

Figure 2 Metopiellus crypticus sp. nov.

Habitus, dorsal view (A); habitus, left lateral view (B); head and pronotum, dorsal view (C); proleg (D); left antenna, lateral view (E); abdomen of male, ventral view (F); abdomen of female, ventral view (G); aedeagus, ventral view (H); aedeagus, lateral view (I); aedeagus, dorsal view (J); left tergum IX (K); right tergum IX (L). Scale bars: one mm (A–B); 0.5 mm (C, E); 0.2 mm (D, F–L). Holotype male (A–F, H–L). Paratype female (G).

Figure 3 Representative genetic map of the mitogenome of Metopiellus crypticus sp. nov.

Disposition of all 37 mitochondrial genes. Colored arrows pointing to the left and right represent the transcription regions of protein coding genes (blue), rRNA genes (red) and tRNA genes (purple) on the L and H strands, respectively. The green and brownish bars above the arrows indicate monomorphic and polymorphic nucleotide sites among the three analyzed genomes, respectively.

Figure 4 Mitogenome phylogenetic relationships among Staphylinidae species.

Majority-rule consensus phylogram of the maximum likelihood analysis evidencing the phylogenetic relationships among Staphylinidae species with available mitogenomes in the GenBank database and the three specimens of Metopiellus crypticus sp. nov. sequenced here, indicating their respective subfamily affiliations. Well-supported groups (BS ≥ 70) are indicated by their bootstrap values near the branches.

urn:lsid:zoobank.org:act:EE11C828-CCDD-4FDC-B090-A3CC80A1A2E1

Type material (seven males, two females). Holotype: BRAZIL, male, labeled “BRAZIL: Pará, /Curionópolis, Serra/Leste, 22M, 650137mE, /9339970mN, WGS84, [5°58′10.59″S, 49°38′36.86″W]/25.iv[April].2017, BioEspeleo leg”.; “Hay-bait trap, /Transect: T2, /Quadrant: E, Parcel: d”; “HOLOTYPE ♂ [red label]/Metopiellus/crypticus sp. nov./Desig. Asenjo et al. 2023” (CEMT-00120424). Paratype: (six males, two females), labeled: “BRAZIL: Pará, /Curionópolis, Serra/Leste, 22M, 652136mE, /9339073mN, WGS84, [5°8′39.63″S, 49°37′31.79″W]/26.iv[April].2017, BioEspeleo leg”.; “Soil sampling, /Transect: T3, /Quadrant: C, Parcel:-” (one male, ISLA-103823). “BRAZIL: Pará, /Curionópolis, Serra/Leste, 22M, 650360mE, /9339477mN, WGS84, [5°58′26.62″S, 49°38′29.57″W]/25.iv[April].2017, BioEspeleo leg”.; “Soil sampling, /Transect: T1, /Quadrant: D, Parcel:-” (one female, ISLA-103824). “BRAZIL: Pará, /Curionópolis, Serra/Leste, 22M, 652013mE, /9339211mN, WGS84, [5°58′35.15″S, 49°37′35.80″W]/26.iv[April].2017, BioEspeleo leg”.; “Soil sampling, /Transect: T4, /Quadrant: C, Parcel:-” (one male, MPEG-01051329). “BRAZIL: Pará, /Curionópolis, Serra/Leste, 22M, 650095mE, /9339732mN, WGS84, [5°58′18.34″S, 49°38′38.21″W]/25.iv[April].2017, BioEspeleo leg”.; “Hay-bait trap, /Transect: T2, /Quadrant: A, Parcel: d” (1 male, MPEG-01051330). “BRAZIL: Pará, /Curionópolis, Serra/Leste, 22M, 650095mE, /9339732mN, WGS84, [5°58′18.34″S, 49°38′38.21″W]/25.iv[April].2017, BioEspeleo leg”.; “Hay-bait trap, /Transect: T2, /Quadrant: A, Parcel: c” (two male, CEMT-00120425 and CEMT-00120426, and one female, CEMT-00120427). “BRAZIL: Pará, /Curionópolis, Serra/Leste, 22M, 650070mE, /9339845mN, WGS84, [5°58′14.67″S, 49°38′39.03″W]/25.iv[April].2017, BioEspeleo leg”.; “Hay-bait trap, /Transect: T2, /Quadrant: C, Parcel: b” (1 male, MPEG-01051331). All paratypes with label “PARATYPE [yellow label]/Metopiellus/crypticus sp. nov./Desig. Asenjo et al. 2023”.

Additional specimens. BRAZIL: Pará, Curionópolis, Serra Leste, 22M, 650137mE, 9339970mN, WGS84, [5°58′10.59″S, 49°38′36.86″W], 25.iv[April].2017, BioEspeleo leg., HBT-T2 E(B) (1 male, ITV10661). BRAZIL: Pará, Curionópolis, Serra Leste, 22M, 650070mE, 9339845mN, WGS84, [5°58′14.67″S, 4938′39.03″W], 25.iv[April].2017, BioEspeleo leg., HBT-T2 C(A), (1 female, ITV21026). BRAZIL: Pará, Curionópolis, Serra Leste, 22M, 650070mE, 9339845mN, WGS84, [5°58′14.67″S, 4938′39.03″W], 25.iv[April].2017, BioEspeleo leg., HBT-T2 C(A), (1 female, ITV21027).

Diagnosis. Among the members of Metopiellus, the new species Metopiellus crypticus Asenjo sp. nov. is very similar to M. painensis because both have a similar habitus (Figs. 2A–2B; Fig. 1 in Asenjo, Ferreira & Zampaulo Rde, 2017) and eyes nearly absent (Figs. 2A–2C; Fig. 3 in Asenjo, Ferreira & Zampaulo Rde, 2017), but the new species differs by having the antennomere 7 rounded (Fig. 2E) (rectangular in Metopiellus painensis [Fig. 5 in Asenjo, Ferreira & Zampaulo Rde, 2017]). Furthermore, Metopiellus crypticus Asenjo sp. nov. further differs by the paramere asymmetric elongate with apex bifurcated (Figs. 2H–2J) (paramere asymmetric not bifurcated in M. painensis [Figs. 10–12 in Asenjo, Ferreira & Zampaulo Rde, 2017]). Also, Metopiellus crypticus sp. nov. differs by the median lobe curved and edge with long line of small teeth (Figs. 2H–2J) (median lobe almost right with the apex curved without small teeth in M. painensis [Figs. 10–12 in Asenjo, Ferreira & Zampaulo Rde, 2017]).

Holotype male, BL: 2.68. Body, mouthparts, antennae and tarsi light brown (Figs. 2A–2B).

Head: pyriform (Figs. 2A and 2C), length (HL: 0.44) similar to width (HW: 0.44), anterior region narrower and ending in an emarginated, antennal tubercle. Posterior margin of head narrowing, with posterolateral angles rounded. Neck almost half of width (NW: 0.19) of head. Head with two vertexal foveae [VF] (Figs. 2A and 2C), foveae connected by a transverse sulcus near posterior margin. Vertex longitudinally impressed, with weak sulcus running from anterior margin of antennal tubercle to neck. Ventral surface of head without gular sulcus and posterior region with two gular foveae [GF] connected by curved sulcus. Eyes (Figs. 2A–2C) composed of some ommatidia situated at middle of head length in lateral view. Antennae (Fig. 2E) almost 3/4 body length, scape almost half antenna length, last three antennomeres gradually broadening. Scape length 0.92 mm, width 0.09 mm, pedicel shorter than scape (0.39: 0.08), antennomere 3 (0.06: 0.06), antennomere 4 (0.05: 0.06), antennomere 5 (0.07: 0.06), antennomere 6 (0.06: 0.06), antennomere 7 (0.06: 0.08), antennomere 8 (0.03: 0.06), antennomere 9 (0.06: 0.1), antennomere 10 (0.06: 0.12), antennomere 11 (0.15: 0.14); all antennomeres covered by long microsetae.

Thorax: pronotum (Figs. 2A, 2C) slightly wider than long (PL: 0.45; PW: 0.52) widest at anterior half. Pronotum convex with weak median longitudinal sulcus, each side with lateral sulcus, with transversal antebasal sulcus. Pronotum with basal and anterior margins weakly emarginated; with median antebasal fovea (MAF) and lateral antebasal fovea (LAF). Prosternum with lateral procoxal fovea (LPCF). Mesoventrite with median mesocoxal fovea (MMNF), lateral mesosternal foveae (LMNF) lateral mesocoxal foveae (LMCF), and with lateral metasternal foveae (LMTF). Metaventrite with median metasternal fovea (MMTF) and one flat median triangle area before metacoxal cavities.

Elytra: subquadrate (EL: 0.74; EW: 0.80), sides gradually broadening apically (Fig. 2A). Posterior margins slightly concave, discal stria (DS) and sutural stria (SS) present. Elytron with two basal elytral foveae (BEF) at anterior margin, one at side of base of the elytral sutural stria, second on the base of discal stria. Apico-lateral margin of elytra with a small notch. Flight wings absent.

Legs: Legs long and slender (Figs. 2A–2B). Femora thickened in apical half. Tibiae curved and similar in length to femora, all tibiae thickened at apex. Protibiae carinate and open at base, lacking microsetae on the concave mesial face (Fig. 2D). Tarsi 3-segmented (Fig. 2D), first tarsomeres short, last 2 tarsomeres longer, tarsomere 2 longer than segment 3; all tarsi with single claw and minute accessory seta. Procoxae conical and prominent, mesocoxae rounded and prominent, metacoxae transverse, region articulated with trochanter conical in shape. Procoxae with small, apically pointed prosternal process, mesocoxae weakly separated, metacoxae contiguous.

Abdomen: strongly margined (Fig. 2A), with five visible tergites (morphological tergites IV-VIII), tergite III reduced to translucent plate beneath elytra, tergite IV with basolateral fovea (BLF), tergite VIII with apex straight. Tergites IV-VII bordered by distinct paratergites, paratergite in abdominal segment IV with one small tooth in the middle. Sternite III with transverse depressed plate completely bare and beneath metacoxae, transverse plate with longitudinally projecting carina at middle. Sternite IV with baselateral fovea (BLF). Tergum IX divided into two plates; right plate (Fig. 2K) larger and more sclerotized than left (Fig. 2J). Sternite VIII (Fig. 2F) with apex deeply emarginate.

Aedeagus (Figs. 2H–2J): asymmetric with paramere partially fused to form elongate plate with apex forked, the median lobe slightly bulbous at base, elongate and narrow, stronger curved laterally at apex, on edge with line of small teeth.

Female. Similar to male, except apex of tergite VIII convex (Fig. 2G).

Distribution. Only known from Serra Leste, Curionópolis, Pará, Brazil (Figs. 1A–1C).

Etymology. The specific epithet “crypticus” is a noun in apposition.

Mitogenome sequence and phylogenetic placement

All three assembled mitogenomes (GenBank accession numbers MZ576843 (ITV10661), MZ576844 (ITV21026) and MZ576845 (ITV21027)) presented the standard structure sequence and gene content for Metazoa, consisting of 13 PCG, 22 tRNA genes and two rRNA genes (Fig. 3). The three mitogenome assemblies ranged in size from 14,353 to 14,984 bp, with similar GC contents between 16.2% and 16.5%, and 98.3% identical sites (1.7% differences). We observed differences in nucleotide composition among the three mitogenomes of Metopiellus crypticus Asenjo sp. nov., with indel events mostly occurring in the rRNA genes (three in each locus). Also, rrnL presented 33 site substitutions as indicated by the mismatches in the alignment, one of the highest proportions of polymorphic sites within the analyzed mitogenomes (2.68% of the 1232 bp), behind only of NAD6 (3.73% of the 456 bp), excluding the tRNAs, which are considerably shorter with 63 bp on average (Table 1).

Table 1 General features of the mitochondrial genes of Metopiellus crypticus sp. nov.

Gene	Size (bp)	Indels	Mismatches	% Mismatches	Coding strand	Start codon	Stop codon	
ATP6	651	0	0	0.00	L	ATG	TAA	
ATP8	153	0	1	0.65	L	ATT	TAA	
COB	1110	0	26	2.34	L	ATA	TAA	
COX1	1537	0	11	0.72	L	ATT	T	
COX2	682	0	0	0.00	L	ATA	T	
COX3	781	0	8	1.02	L	ATG	T	
NAD1	924a	1	18	1.95	H	ATA	TAA	
NAD2	963	0	1	0.10	L	ATA	TAA	
NAD3	348	0	2	0.57	L	ATT	TAG	
NAD4	1332a	0	29	2.18	H	ATG	TAA	
NAD4L	273	0	4	1.47	H	ATT	TAA	
NAD5	1692	0	35	2.07	H	ATT	TAA	
NAD6	456a	1	17	3.73	L	ATT	TAA	
rrnL	1232	3	33	2.68	H	–	–	
rrnS	728a	3	4	0.55	H	–	–	
trnA (tgc)	52	0	0	0.00	L	–	–	
trnC (gca)	63a	1	0	0.00	H	–	–	
trnD (gtc)	64	0	1	1.56	L	–	–	
trnE (ttc)	62	0	0	0.00	L	–	–	
trnF (gaa)	63a	1	1	1.59	H	–	–	
trnG (tcc)	63	0	2	3.17	L	–	–	
trnH (gtg)	63	0	1	1.59	H	–	–	
trnI (gat)	63	0	0	0.00	L	–	–	
trnK (ctt)	68	0	0	0.00	L	–	–	
trnL1 (tag)	61	0	0	0.00	H	–	–	
trnL2 (taa)	62	0	0	0.00	L	–	–	
trnM (cat)	68b	1	0	0.00	L	–	–	
trnN (gtt)	63	0	0	0.00	L	–	–	
trnP (tgg)	64a	1	0	0.00	H	–	–	
trnQ (ttg)	63	0	0	0.00	H	–	–	
trnR (tcg)	60	0	0	0.00	L	–	–	
trnS1 (tct)	55	0	0	0.00	L	–	–	
trnS2 (tga)	64	0	2	3.13	L	–	–	
trnT (tgt)	64	0	0	0.00	L	–	–	
trnV (tac)	64	0	2	3.13	H	–	–	
trnW (tca)	64a	1	0	0.00	L	–	–	
trnY (gta)	63	0	1	1.59	H	–	–	
Notes.

Sequenced mitogenomes based on from three different specimens, indicating the size of the transcription regions, presence of indel events, number of mismatches after the alignment of the mitogenomes, coding strand, and sequences of both start and stop codons.

a For genes with indel events, we presented the length observed in two of the three specimens.

b For trnM (cat), the value presented correspond to the average size among the three mitogenomes.

Most genes were encoded in the L-strand, including nine PCGs (ATP6, APT8, COB, COX1, COX2, COX3, NAD2, NAD3 and NAD6) and 14 tRNA genes (trnA, trnD, trnE, trnG, trnI, trnK, trnL2, trnM, trnN, trnR, trnS1, trnS2, trnT and trnW). Also, ATT was the most frequent start codon, being observed in seven genes, followed by ATA in four genes, and ATG in three (Table 1). On the other hand, almost all genes presented the TAA stop codon, except for NAD3 with TAG, and the three COX genes with an incomplete stop codon (Table 1).

Previously published mitogenome sequences of Staphylinidae species from 11 subfamilies, plus one species of Hydrophilidae (Cercyon borealis) and one of Histeridae (Euspilotus scissus) to be used as outgroups, were obtained from GenBank, totaling 61 accessions. Sequences of the 13 protein coding genes (PCG) were aligned with MAFFT v7.45 (Katoh et al., 2002; Data S3) and maximum likelihood (ML) phylogenetic trees were obtained using RAxML v8 (Stamatakis, 2014), implemented in raxmlGUI v2 (Edler et al., 2021) using the model GTR+PROTGAMMA and the rapid bootstrapping option with 1,000 replicates (Data S4).

In the phylogenetic analysis, most of the subfamilies were recovered as monophyletic and well supported, except for Tachyporinae and Paederinae, which were polyphyletic and paraphyletic, respectively, and Staphylininae, presenting a low bootstrap support (BS = 55) (Fig. 4). Within Pselaphinae, the relationships among the sampled species were mostly unsupported (BS <70). The three specimens of Metopiellus crypticus sp. nov. grouped with maximum statistical support (BS = 100) in the longer branch within the subfamily, being recovered as sister to Batrisodes sp., although with low statistical support (BS = 42) (Fig. 4).

Discussion

The new species belongs to the genus Metopiellus based on the third antennal segment being much shorter than the second (Fig. 2E); the posterior coxae contiguous or nearly so; and the mesial face of protibia being carinate and open at its base and apex (Fig. 2B) (Raffray, 1908; Park, 1942; Asenjo, Ferreira & Zampaulo Rde, 2017). One of the characters “pronotum not being spinose” for Metopiellus, should not be considered a good character as considerated by previous authors to define the genus since M. guanano has pronotum with four small spines (Fiorentino, Tocora & Ramirez, 2022).

Specimens of the known species on the genus Metopiellus were collected in litter of ants or in caves (Asenjo, Ferreira & Zampaulo Rde, 2017). Unlike the other species of the genus, which have been recorded in forested areas or inside caves, the new species has been found in forested areas of the Serra dos Carajás, as well as in a savanna-like environment, although being less abundant in the latter.

For the first time, the mitochondrial genome of a Metopiellus species is described focusing on the phylogenetic position of Metopiellus crypticus sp. nov. within the subfamily Pselaphinae. In the obtained topology, the new species was recovered as sister to Batrisodes sp., although with low statistical support (BS = 42). However, this grouping was probably an artefact influenced by the absence of published mitogenomes of the others representatives of the other Metopiasini.

Despite of all three assembled mitogenomes presenting the standard structure sequence and gene content for Metazoa, we could not obtain a circularized assembly for any of them, probably due to a high repetitive DNA content in the D-loop control region (Sayadi et al., 2017), as indicated by the several mononucleotide repeats in both ends of the assembled sequences. Such a pattern has been frequently reported for beetle species, with several Coleoptera mitogenomes available in the GenBank database containing all expected genes, but missing part of the control region, and thus being reported as partial sequences.

Supplemental Information

Supplemental Information 1 Mitogenome of Metopiellus crypticus, accession no. ITV10661, GenBank no. MZ576843

Click here for additional data file.

Supplemental Information 2 Mitogenome of Metopiellus crypticus, accession no. ITV21026, GenBank no. MZ576844

Click here for additional data file.

Supplemental Information 3 Mitogenome of Metopiellus crypticus, accession no. ITV21027, GenBank no. MZ576845

Click here for additional data file.

Data S1 Bash script for cleaning Illumina reads and assembling mitogenomes

Click here for additional data file.

Data S2 Metopiellus crypticus sp. nov. measurements, comparing the holotype specimen with eight paratypes

Click here for additional data file.

Data S3 Amino acid sequence matrix of Staphylinidae species using the 13 protein coding genes of the sampled mitogenomes

Click here for additional data file.

Data S4 Commands used to run the maximum likelihood analysis in RAxML

Click here for additional data file.

Additional Information and Declarations

Competing Interests

Author Contributions

Field Study Permissions

DNA Deposition

Data Availability

New Species Registration

Guilherme Oliveira is an Academic Editor for PeerJ. Marcus Paulo Alves de Oliveira is employed by BioEspeleo Consultoria.

Angélico Asenjo conceived and designed the experiments, performed the experiments, analyzed the data, prepared figures and/or tables, authored or reviewed drafts of the article, and approved the final draft.

Marcus Paulo Alves de Oliveira conceived and designed the experiments, authored or reviewed drafts of the article, and approved the final draft.

Renato R.M. Oliveira performed the experiments, analyzed the data, authored or reviewed drafts of the article, and approved the final draft.

Eder Soares Pires performed the experiments, authored or reviewed drafts of the article, and approved the final draft.

Marcely Valois analyzed the data, authored or reviewed drafts of the article, and approved the final draft.

Guilherme Oliveira conceived and designed the experiments, authored or reviewed drafts of the article, and approved the final draft.

Santelmo Vasconcelos conceived and designed the experiments, performed the experiments, analyzed the data, prepared figures and/or tables, authored or reviewed drafts of the article, and approved the final draft.

The following information was supplied relating to field study approvals (i.e., approving body and any reference numbers):

The samples were collected under the sampling permit 49.994, as granted by ICMBio/MMA.

The following information was supplied regarding the deposition of DNA sequences:

The assembled mitogenomes are available at GenBank: MZ576843, MZ576844 and MZ576845.

The following information was supplied regarding data availability:

The troglogen data is available at NCBI: PRJNA862473.

The following information was supplied regarding the registration of a newly described species:

Publication LSID: urn:lsid:zoobank.org:pub:85F589F2-AECD-4D52-B545-6363C4CE8E18

Metopiellus crypticus: urn:lsid:zoobank.org:act:EE11C828-CCDD-4FDC-B090-A3CC80A1A2E1

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
