# Peer review of "The complete mitochondrial genome and description of a new cryptic Brazilian species of Metopiellus Raffray (Coleoptera: Staphylinidae: Pselaphinae)"

_PeerJ, doi:10.7717/peerj.15697_

## Round 0.1 · original submission · Minor Revisions

Overall a thorough article and worthy of publication. More detail is needed in methods about settings and parameters used in the various software packages used for molecular analyses. The article needs some attention to English language editing. Reviewers have provided some editing suggestions in their marked-up PDF files so please incorporate these edits.

Reviewer 1 ·

Basic reporting

The manuscript language is clear; literature references are cited according to the current knowledge in the field, and figures are well-organized and prepared. The presented results are based on a sufficient amount of data.

Experimental design

Methods are described in detail, the knowledge gaps are clearly stated, and the research has been performed according to the standards in the field.

Validity of the findings

All data on the collected species are provided along with the genomic results. The authors describe a new species and provide the first mitogenome for the Pselaphinae tribe. Data are publicly available.

Additional comments

The manuscript will be a valuable addition to our knowledge, and I strongly recommend it for publication. There are only some minor comments/corrections that I marked directly in the file (attached).

Annotated reviews are not available for download in order to protect the identity of reviewers who chose to remain anonymous.

·

Basic reporting

The manuscript describes a new species of Pselaphine rove beetles and provides 3 mitogenomes of this species, which are the first to be produced for the tribe (Metopiasini). The paper is generally well written with a couple of minor unambiguous parts that I have highlighted. The reference, figures and tables are good and appropriate. The raw data is shared and deposited in open access online repository.

Experimental design

Firstly the paper goes on to describe the new species, which is nicely done except for some minor comments added in the pdf. After it explains how 3 mitogenomes were produced, which again is good. My only comment here is to better report the library preparation steps, so that it is clear whether the three samples were indexed and pooled at the sequences stage. Lastly it goes on to perform a phylogenetic analysis with the available mitogenome for the whole family. I don't really see the use of this analysis, as it only places the genus in Pselaphinae, which is already pretty clear. Instead I would have looked at COI or other markers found within mitogenomes, where more data is present. Perhaps such approach could be added to shed more light on the placement of the species/genus.

Validity of the findings

Species and their description is still lacklustre in many parts of the world, especially in species rich tropical region. Therefore it is always nice to see additional species being well described. An issue in these regions is the little reference genetic data available. This is also clear here, where nothing was matching the mitogenomes of the newly described species. With the open access publication of the data it opens for the possibility of others getting a match in the future, so a clear step in the right direction.

Additional comments

* There are a few issues with the language, I have tried to comment/edit where this is happens.
* Please add a little more details about the library preparation steps. Especially whether samples were sequenced together through indexing.
* Geoereferencing is given in UTM coordinates. I would suggest to also add decimal degrees, as this makes it much easier to enter into various mapping sources.
* In the diagnosis it is not quite clear how the new species is distinguished from other Metopiellus species that are not M. crypticus.
* In the diagnosis it is also not quite clear how this species is clearly part of the genus Metopiellus, this is discussed in the discussion, but a short explanation could be added to diagnosis as well.
* It looks like not much genetic data is matching these newly generated mitogenomes. I would add some such on COI barcodes or other relevant marker in GenBank BLAST (https://blast.ncbi.nlm.nih.gov/Blast.cgi) and BOLD ID engine (https://www.boldsystems.org/index.php/IDS_OpenIdEngine) , to see if you could get better matches that could be used . It may not be that you get matches, but this is interesting as well, as it may represent very unique genetic data, such result should be highlighted.

Reviewer 3 ·

Basic reporting

The structure of the article is correct. The language is understandable. The figures are informative, but one is not quoted in the text.
In addition, I suggest adding molecular alignment as supplement material.
All my comments are added in pdf.

Experimental design

For better repeatability of analyses, please attach a script with all steps.

Validity of the findings

No comment

Annotated reviews are not available for download in order to protect the identity of reviewers who chose to remain anonymous.

---

## Round 0.2 · accepted · Accept

As all of the reviewers' concerns have been addressed, the manuscript is now acceptable for publication.

---

## Author Rebuttal · Round 0.2

**To**
**Dr. Andrew Mitchell**
**PeerJ, Academic Editor**

Dear Editor,

We are grateful for the reviews we received, which undoubtedly will contribute to enhancing our work. The manuscript was modified according to the comments of the reviewers, and all featured comments are listed and answered below.

**REVIEWER 1**

**Line 155:**
Comments: _Maybe better to say "with apex ...."_ and _Maybe better "not bifurcated"_
- ✓ We have changed the text accordingly.

**Line 158:**
Comment: _Does M. painensis also have teeth on paramere? If not, it would be important to mention it here as well._
- ✓ We have changed the text to clarify the sentence.

**Line 168:**
Comment: _It should be "ommatidia" if you mean there is more than one._
- ✓ We have changed the text accordingly.

**Line 171:** _I am not sure what this refers to? It looks like it should be in another place._
- ✓ To clarify the sentence, the text was changed, which also covering the comments made by the Reviewer #3.

**Line 191:**
Comment: _This should be reworded or moved, e.g. "...carinate and open at base mesial face"_
- ✓ Done.

**Line 192:**
Comment: _How to distinguish "very short" from "short"?_
- ✓ We deleted the qualifier "very" in the sentence.

**Line 205:**
Comment: _In the diagnosis it was said there is only one paramere. Please, clarify or change the wording in diagnosis._
- ✓ We modified the text accordingly.

**Line 209:**
Comment: _But the specimens were collected in different sites, right? Please, clarify what you mean by the single type locality here._
- ✓ We modified the text to cover the total sampling area of the new species.

**Figure 4:**
Comment: _Should be Paederinae_
- ✓ The name was corrected in the modified figure file.
* * *
**REVIEWER 2**

**Line 22:**
Comment: _It is also worth specifying the year_
- ✓ Done, we added the year the species were described.

**Line 24:** _If there is only one author of the name, his surname should appear from the beginning: Asenjo sp. nov._
- ✓ Done.

**Line 26:**
Comment: _Every time a new species is mentioned it should have "sp. nov." indications_
- ✓ Done.

**Line 67:**
Comment: _I highly recommend adding a script showing all the steps taken in all analyses. It will allow not only for repetition but will be helpful for beginners._
- ✓ We have added a file containing the scripts for the commands, as indicated in the text.

**Line 161:**
Comment: _The description should not contain any articles_
- ✓ Done, we verified the verified the manuscript and corrected accordingly.

**Lines 233 and 234:**
Comments: _I suggest adding the obtained alignment to the supplemental material_ and _It is good to give the full command or all parameters used in the analysis_
- ✓ We have provided the alignment and the command for the RAxML analysis as Supplementary Data S3 and S4, respectively, as indicated in the text.

**Line 362:**
Comment: _Figure 1C is not cited anywhere in the text_
- ✓ We corrected the mention in the line 203 – it was: **Figs. 1A-B**, and now it is **Figs. 1A-C**.
* * *
**REVIEWER 3**

**Line 56:**
Comment: _isn't litter sampling a litter-associated collecting method?_
- ✓ We adjusted the text, indicating that it was "soil sampling by flotation".

**Line 65:**
Comment: _Add a little more information on the library preparation. From what i can read each samples would be dual indexed, so they could be sequenced with other samples. If this correct? If yes, then add this information, otherwise write how you were able to obtain 3 mitogenomes from one sequencing run._
- ✓ We described in more detail the used methods, covering the points indicate by the reviewer.

**Lines 68 and 69:**
Comment: _why is at least added here? Do you expect that there are even more reads. You should know the total number. If this refers to the lowest number of reads for each mitogenome, then i would report the number of reads for each mitogenome seperately._
- ✓ We added the information for all three mitogenomes in the text.

**Line 80:**
Comment: _Why not do measurements on multiple specimen to show variation?_
- ✓ We performed measurements for additional specimens, with the data being presented in the Supplementary Data S2.

**Line 84:**
Comment: _in mm? Or what scale?_
- ✓ We mentioned the used scale in the previous paragraph.

**Line 124:**
_Comment: I would add as an explanator the decimal degrees of the coordinates. In this case: [-49.6435E]. Looks like the following converter can be used: https://sigam.ambiente.sp.gov.br/sigam3/Controles/latlongutm.htm?latTxt=ctl00_con_
- ✓ Done. We added the locations in decimal degrees.

**Line 143:**
Comment: _Why are these specimen not part of the types series as paratypes?_
- ✓ Due to the small size of the specimens, these samples were used (and therefore destroyed) in the DNA extraction procedures.

**Line 158:**
Comment: *Perhaps it would be good with a short comment on how it is distinguished from other Metopiellus speices that are not M. crypticus.*
- ✓ We modified the text to clarify the sentence.

**Line 162:**
Comment: *No quite sure what is meant here.*
- ✓ We modified the text to clarify the sentence.

**Lines 170 and 171:**
Comment: *Move this explanation to methods section and reword. Now it is not quite clear, but my guess it would be: maximum length of antennomere without peduncle in mm.*
- ✓ Done.

In addition to the corrections indicated by the reviewers, we also included a few other modifications, such as the requirements such as the technical changes required regarding the LSID for the new species and the authorship of the Figure 1, which are also highlighted in the text, aiming to improve the manuscript.

Thank you very much for your kind attention.

Sincerely yours,

**Santelmo Vasconcelos**
Instituto Tecnológico Vale D.S.
Rua Boaventura da Silva 955
66055-090 Belém, Pará, Brazil
Phone: +55 91 3213 5400
santelmo.vasconcelos@itv.org